# The Combined Influence of Cover Crops and Manure on Maize and Soybean Yield in a Kentucky Silt Loam Soil

**Maheteme Gebremedhin [1],\* , Sait Sarr [2], Mark Coyne [3], Karamat R. Sistani [4] and Jason Simmons [4]**

1   College of Agriculture, Communities, and the Environment, Kentucky State University, Frankfort, KY 40601, USA

2   School of Urban and Public Affairs, University of Louisville, Louisville, KY 40208, USA; sait.sarr@louisville.edu

3   Department of Plant and Soil Sciences, University of Kentucky, Lexington, KY 40546, USA; mark.coyne@uky.edu

4   USDA-ARS, Food Animal Environmental Systems Research Unit, Bowling Green, KY 42104, USA; karamat.sistani@ars.usda.gov (K.R.S.); jason.simmons@ars.usda.gov (J.S.)

\*   Correspondence: maheteme.gebremedhin@kysu.edu; Tel.: +1-502-597-6830

**Abstract:** Management that degrades soil can be one of the main causes of low agricultural productivity and environmental problems in many agricultural regions. There is renewed interest in soil conservation practices to promote sustainable agriculture by improving soil quality and productivity. In this study, the short-term on-farm benefits of cover crops and manure on crop yield and biomass were examined during two consecutive growing seasons. The experiment was conducted at a small-producer farm in Logan County, Kentucky, USA. Soybean (*Glycine max* L.) and maize (*Zea mays* L.) were used as summer annual rotation crops in no-tilled soils. a cover crop mix of cereal rye (*Secale cereale* L.), Austrian winter pea (*Pisum sativum* L.), and crimson clover (*Trifolium incarnatum* L.) was planted after the main crop was harvested each year. Aboveground biomass of the soybean and maize were assessed, and yield was estimated from hand-harvested plants. In the first year of the study (2016), there were apparent but not significant beneficial effects of animal manure and cover crops on soybean yield, but not on biomass. The biomass and maize grain yield in the second year (2017) were detectable, significant, and increased as a result of the cover crops and manure application ($p < 0.05$). While beneficial effects of combining cover crops and manure may not be obvious in the first year of a rotation, they can be apparent in subsequent years. However, longer-term observation and measurement are necessary to better quantify the relationship between sustainable conservation practices and productivity.

**Keywords:** biomass; conservation practices; cover crops; manure; soil quality; yield

## 1. Introduction

Soil management, sustainability, and productivity are a global concern for food security and environmental quality [1,2]. Agricultural intensification has caused soil compaction, organic carbon loss, increased evaporation rates, excessive nutrient runoff, and biodiversity loss, all negatively affecting agricultural soil productivity, sustainability, and resilience [2]. Soil conservation practices such as cover cropping and amendment with animal manure may mitigate effects of soil degradation and hold the promise of increasing soil quality, crop productivity, and ultimately decreasing soil degradation [2,3].

Cover crops provide several agronomic and environmental benefits, such as protection from soil erosion [3], increased soil organic matter, soil porosity, and water infiltration, improved soil structure, and contribute to efficient nutrient cycling [4]. In a crimped and rolled system, for example, cover crop residues (i.e., winter rye [*Secale* spp.]) can inhibit early season weed germination and create a path for the subsequent grain crop to outgrow growth- and yield-limiting weeds [5,6] before the rye is ultimately decomposed and mixes with the underlying soil. Cover crops also decrease soil compaction [7,8], prevent phosphorus (P) loss, and take up residual nitrate ($NO_3^-$) [9–11]. Cover crops like oats (*Avena sativa*) and rye (*Secale* spp.) reduce potential $NO_3^-$ leaching in maize - soybean rotations [12,13]. Adverse effects of cover crops, which may inhibit their adoption, include N immobilization [14], competition for nutrients and water [15], and promotion of pest animals (e.g., voles *[Microtus ochragaster])* [16].

Manure, as a source of nitrogen (N) and P, can improve soil quality and increase maize yield compared to chemical fertilizers [17]. Several studies [18,19] support the advantage of using animal manure rather than chemical fertilizers. Significant increase in soil organic matter and wheat (*Triticum aestivum*) yield was observed with manure-treated plots compared to plots treated with chemical fertilizers [18]. Similarly, an increase in soybean grain yield was observed in manure-amended plots compared to control plots and plots treated with chemical fertilizers [19]. In addition, manure adds organic matter to the soil [20], which may act as a soil binding agent [21] and promotes N transformation and nutrient cycling [22].

Cover crops and animal manure increase soil mineralizable N—the amount of soil N available to crops during a growing season [23]. This will offset some chemical fertilizer N use in following seasons [24]. For example, fertility recommendations from the University of Kentucky Cooperative Extension Service give up to a 28 kg ha$^{-1}$ credit to maize grown following four or fewer years of a leguminous cover crop [25]. Compared to chemical fertilizers, manure mineralization provides similar nutrient forms [26]. Previous studies suggest winter legume cover crops release 50 and 150 kg N ha$^{-1}$ as plant-available N [27,28]. Combining the two—manure and legume N cover crops—may be necessary to fulfill some crop (maize) N requirements during the growing season of application because the N from the legume cover crops alone may not satisfy maize demands in a no-till system [29,30].

A 4-year study in Western Kentucky, USA, to find a best management practice integrating crop rotation, animal manure use, and winter cover crops to improve soil quality and crop yield, indicated that maize grain yield without cover crops decreased by 5% while plots with cover crops produced about a 5% yield increase [31]. Nkongolo and Haruna [32] assessed the effects of crop rotation and cover crops on maize and soybean yields from 2011 to 2013 and found that soybean yield increased significantly with cover crops in 2011 and 2012. In contrast, maize yield was not affected. These results suggest that cover crops may have a significant, if contradictory, influence on crop yield.

The number of small-scale farmers embracing conservation practices in the US has been modest over the years [33] although an increasing number of farmers are beginning to realize the environmental benefits of these practices. The low adoption rate could include: (1) few mechanisms to incorporate research on manure use or cover cropping practices into farm level decisions; (2) proof of short-term benefits for conservation practices on yield [34]. As more producers in the US incorporate cover crops into their rotation, the expectation is that they will see the benefits of this conservation practice with improvements in soil and crop performance with time. Our goal in this study was to show an immediate, additive effect of conservation practices that included both cover crops and animal manure on maize and soybean yield and biomass that was greater than a simple rotation effect.

## 2. Materials and Methods

### 2.1. Research Site, Cropping System, and Experiment Design

This study was performed in Logan County, Kentucky, USA (36.88°N; 86.60°W). The site has a warm humid temperate climate with hot summers. Very cold periods in winter are usually of short

duration. Mean annual temperature varied between 7 ◦C in January to 21 ◦C in July. Mean annual precipitation varied from 991 mm in 2015 to 1524 mm in 2016 [35]. The soil is a well-drained Crider silt loam (fine-silty, mixed, active, mesic Typic Paleudalfs) according to USDA soil taxonomy. In general, the area is considered prime farmland. The site had been in a no-till maize-soybean rotation for approximately 10 years (personal communication with the landowner) before the start of this study. Initial soil physical, chemical, and biological properties from 0 to 15 cm depth are in Table 1.

**Table 1.** Soil physical, chemical, and biological properties (0–15 cm depth) prior to initiating the study: bulk density (BD), organic matter (OM), total nitrogen (N), phosphorus (P), potassium (K), cation exchange capacity (CEC), and potentially mineralizable soil N (MinN).

| BD (g cm$^{-3}$) | Soil pH | OM (%) | N (%) | P (mg kg$^{-1}$) | K (mg kg$^{-1}$) | CEC (meq/100g soil) | MinN (mg kg$^{-1}$) |
|---|---|---|---|---|---|---|---|
| 1.3 | 5.4 | 1.5 | 0.1 | 19 | 142 | 11.3 | 3.7 |

Two crops, maize and soybeans, were grown in a 2-yr rotation starting in 2015 when winter cover crops were planted following maize harvest. The subsequent crop sequence was: soybean–cover crop–maize–cover crop. The cover crop was the same each year: a mix of cereal rye, Austrian winter pea, and crimson clover in which each species was equally represented. Management details are in Table 2.

**Table 2.** Description of the rotation experiment, including dates of termination of cover crops, and planting and harvesting of main crops (soybean and maize) during the study period.

| | 2015–2016 Season (Soybean) | 2016–2017 Season (Maize) |
|---|---|---|
| Fall Cover Crops | Mixture of cereal rye, Austrian winter pea, and crimson clover | Mixture of cereal rye, Austrian winter pea, and crimson clover |
| Cover Crop Planting Dates | 15 October 2015 | 27 October 2016 |
| Cover Crop Termination Dates | 5 May 2016 | 5 May 2017 |
| Main Crop Planting Dates | 25 May 2016 | 7 June 2017 |
| Harvest Dates | 26 October 2016 | 25 October 2017 |

A randomized complete block design (RCBD) with four replications (blocks) was used for the field experiment. The four blocks were separated from each other by 3-meter buffers. Within each block, six treatments were randomly applied for a total of 24 experimental plots: (1) control; (2) cover crops (Cc); (3) manure; (4) manure and cover crops (Manure + Cc); (5) fertilizer (NPK); (6) fertilizer and cover crops (NPK + Cc). Each plot was 6.1 m wide x 18.3 m long. Half of the plots were covered with a mix of three cover crops (Austrian peas, cereal rye, and crimson clover) at a seeding rate of 25 kg ha$^{-1}$ (all species equally represented). Poultry litter, as the manure treatment, was broadcast applied on an N-rate basis to supply approximately 25 kg N ha$^{-1}$ to soybean and 224 kg N ha$^{-1}$ to maize on a plant available basis. Chemical fertilizers consisting of the synthetic compounds were also applied to supply 25 or 224 kg N ha$^{-1}$ to soybean and maize, respectively in the absence of poultry litter. The N application rate and other fertility rates were taken from established recommendations provided by the University of Kentucky's Cooperative Extension Service [25].

*2.2. Biomass Assessment and Yield Measurement/Estimation*

Aboveground biomass of the soybean planted in the first growing season (2016) was assessed on 14 September 2016, using the quadrant method with a harvested area of 0.42 m$^2$. a total of 24 samples were collected (1 sample from each plot), placed in paper bags, labeled, and transported to the Western Kentucky University research farm in Bowling Green KY and dried to constant weight at 65 °C. Biomass measurement of the maize planted in the second growing season (2017) was assessed on

10 October 2017, using a similar procedure to the soybean biomass assessment except that the samples were transported to the Kentucky State University farm in Frankfort KY for drying. The biomass was calculated by extrapolating dry weight values to a per hectare basis (10,000 m$^2$ = 1 hectare).

Soybean and maize grain yield were estimated from hand-harvested plants. Yield was determined by weighing the grain from each plot. The harvested area for soybean was 0.58 m$^2$. Soybean grain yield was adjusted (13%) for a bushel (27.2 kg) to compensate for moisture content, and maize grain yield was adjusted (15.5%) for a bushel (25.5 kg) to compensate for moisture content. The yield was calculated by extrapolating hand-harvest values to a per hectare basis.

### 2.3. Statistical Analysis

Treatment effects on crop biomass and grain yield were examined by one-way ANOVA to compare means of the treatments based on Tukey's test (a single-step, multiple comparison statistical test to find means that are significantly different from each other). ANOVA was performed to see effects of each of the treatments. Analysis was performed using SPSS (IBM SPSS Statistics version 22). We assessed the data for normality ($p = 0.875$) following the Kolmogorov–Smirnov (K-S) test. a significant F test was declared at $p \leq 0.05$. Because of the different crop each year, the data were analyzed separately for each year.

## 3. Results and Discussion

### 3.1. Biomass Assessment

In 2016, when soybean was the main crop, differences in biomass between the treatments were not statistically detectable ($p > 0.05$). The control had the highest total biomass (17.6 Mg ha$^{-1}$) compared to other treatments (Figure 1). This could be because soybean is already a nitrogen fixer and the other key fertility elements (P and K; Table 1) were near optimum according to fertility recommendations [25]. There is also the potential for residual fertility enrichment from the previous season maize stover [36]. In a dry spring, an existing cover crop can also deplete soil water and interfere with germination and growth of a subsequent crop [15]. Another contributing factor could also be that the timing of biomass sampling occurred late in growth.

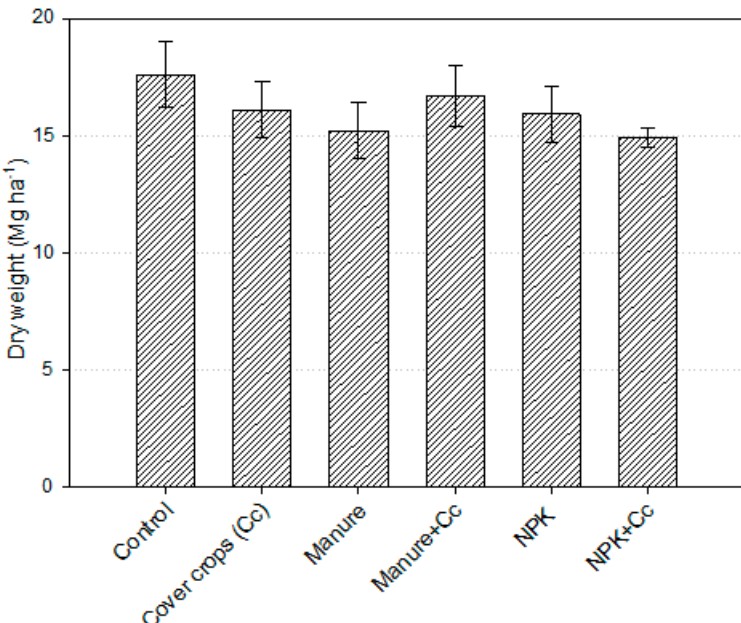

**Figure 1.** Treatment effects of manure and chemical fertilizers with or without cover crops (and an unamended control) on biomass (Mg ha$^{-1}$) for soybean assessed in 2016. Error bars represent standard error of the mean (SEM).

In 2017, when maize was the main crop, treatment differences in biomass were significant ($p < 0.05$). Simply having cover crops improved yield relative to the control (Figure 2). The greatest biomass occurred with the NPK + Cc treatment (23.6 Mg ha$^{-1}$), followed by the Manure + Cc treatment (21.6 Mg ha$^{-1}$) (Figure 2). The biomass of each treatment with a cover crop was at least 8.5 Mg ha$^{-1}$ greater than the control (11.5 Mg ha$^{-1}$). Overall, the combined effects of cover crop and manure use on maize biomass accumulation in 2017 showed a promising direction relative to the control (Figure 2). The interaction of manure + cover crop for maize biomass was both positive and significant. Our initial hypothesis was that the combination of conservation practices would have an additive effect for both biomass and yield. While not purely additive, this result indicates that there is benefit to cover crop and manure use together beyond their utility when used alone for maize.

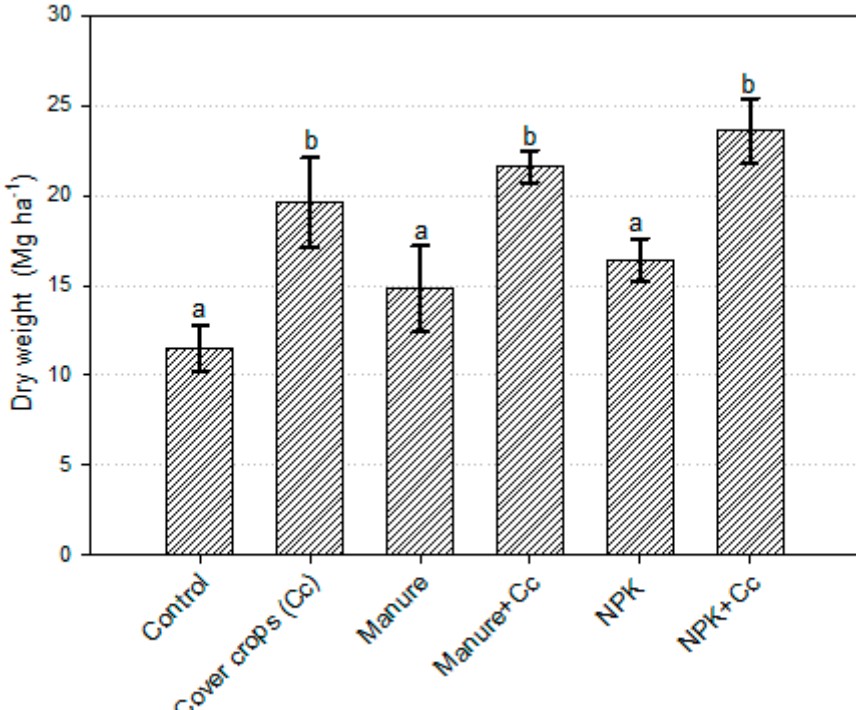

**Figure 2.** Treatment effects of manure and chemical fertilizers with or without cover crop (and an unamended control) on biomass (Mg ha$^{-1}$) for maize assessed in 2017. Error bars represent standard error of the mean (SEM). Different letters denote significant differences among treatments p ≤ 0.05.

### 3.2. Soybean Grain Yield (2016)

There were no statistically detectable differences among treatments ($p > 0.05$) although manure and fertilizer treatments with cover crops trended consistently higher than their non cover crop counterparts (Figure 3). The soybean grain yield in the Manure + Cc treatment was highest (5.6 Mg ha$^{-1}$) among treatments (Figure 3). Yields of 4.5–5.5 Mg ha$^{-1}$ (roughly 67–82 bu/A soybean) are a good field average for this soil environment (E. Ritchey, UK Cooperative Extension, personal communication). While some studies have shown a modest improvement in soybean yield, others have shown unchanged yields [37] from cover crop use. Similar to our work, Olson et al. [38] found no statistically significant differences in soybean yield in no-till plots with and without cover crops throughout a 12-year study in southern Illinois USA. The Olsen study illustrates the problem with tying benefits of conservation practices solely to yield because in the same period there were significant increase in C storage in the slightly eroded soils [38].

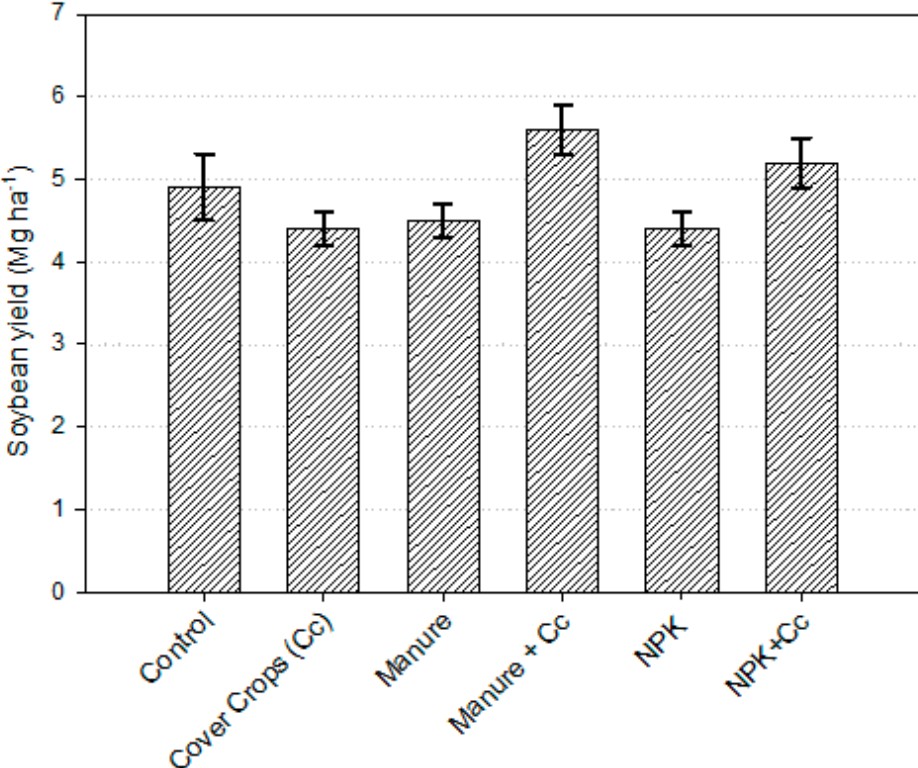

**Figure 3.** Treatment effects of manure and chemical fertilizers with or without cover crop (and an unamended control) on soybean grain yield (Mg ha$^{-1}$) for soybean assessed in 2016. Error bars represent standard error of the mean (SEM).

### 3.3. Maize Grain Yield (2017)

The Manure + Cc treatment was significantly greater than the control and the manure-only treatment ($p < 0.05$) (Figure 4). The yield from the Manure + Cc treatment doubled the yield observed with the control (8 Mg ha$^{-1}$). There was a significant positive interaction between the cover crop and manure treatment, and significant treatment differences were detectable ($p < 0.05$). All cover crop treatments yielded significantly more than the corresponding treatment without cover crops. In a similar study, Sistani et al. [31] found that maize grain yield from treatments with cover crops was about 5% than treatments without cover crops.

Maize grain yield was more affected by cover crop and manure than soybean yield. Yield differences were most likely due to maize response to N as a result of the cover crops, manure, and chemical fertilizer application. Cover crops reduce nutrient losses, take up excess N from the soil during the winter periods, and hence improve soil health and productivity [39]. Because of its responsiveness to N fertility, maize biomass and yield do not behave the same as soybean. Prior work at this site showed a much stronger response of maize than soybean yield to mineralizable N [40]. Some studies have reported benefits from second year use of cover crops for maize, although other cover crops studies have reported minor losses to minor increases in maize yield [37].

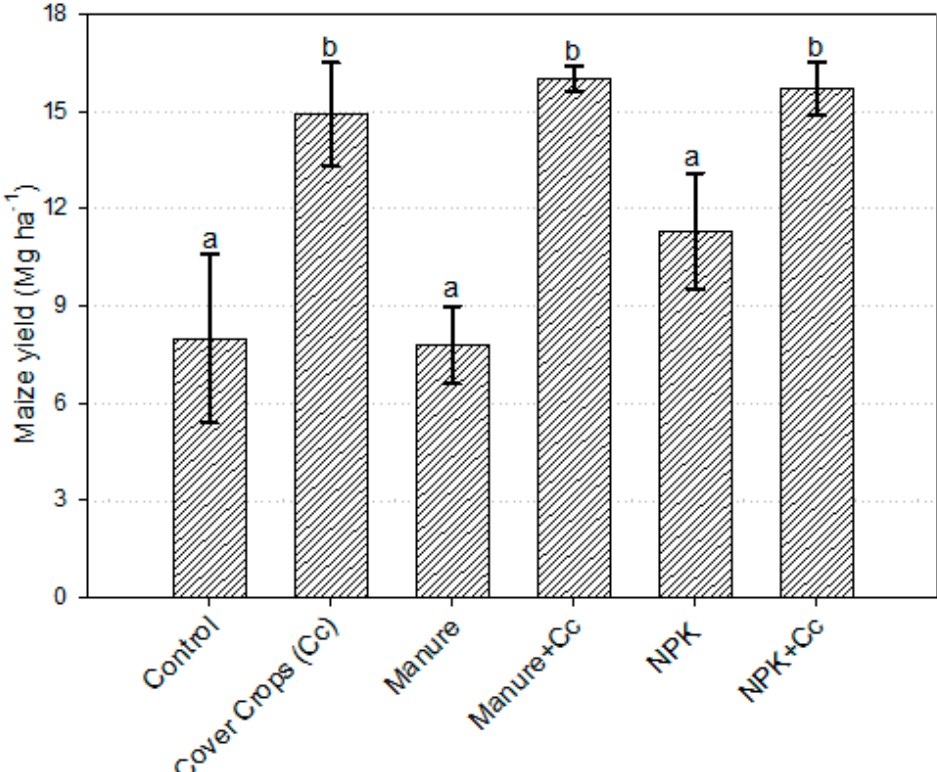

**Figure 4.** Treatment effects of manure and chemical fertilizers with or without cover crop (and an unamended control) on maize grain yield (Mg ha$^{-1}$) for maize assessed in 2017. Error bars represent standard error of the mean (SEM). Different letters denote significant differences among treatments at $p$ ≤ 0.05.

## 4. Conclusions

In the first year of the study (2016), there were no significant combined effects of manure and cover crops on soybean yield and biomass. We can speculate that because fertilization was N-based, and legumes like soybean can fix their own N, they may have received little additional benefit from the winter cover crop mineralization. Alternatively, for that very reason, if the extent of cover crop and soybean N-fixation were overestimated, the crop may have been nutrient limited. However, the biomass and grain yield differences in maize in the second year (2017) showed significant increases because of the cover crops and manure application. Even in the absence of manure, the NPK+Cc treatment showed the cover crop benefit.

Soybean yields were unchanged with cover crops in the first year. However, maize yield and biomass increase was observed in the second year of a rotation that started including cover crops, which could presumably reflect improvements in soil quality and fertility. There is a further need to evaluate the effects of cover crops and manure N application rates on corn and soybean yield, and evaluation of crop and manure-specific effects.

**Author Contributions:** S.S, M.G., K.R.S., and J.S. designed the experiment and collected the data. S.S., M.G., and M.C. analyzed the data and drafted the manuscript.

**Funding:** This study was fully supported by the United States Department of Agriculture-NRCS Conservation Innovation Grant # 68-5C16-15-5251. S. Sarr was supported by a College of Agriculture, Communities, and the Environment Graduate Assistantship.

**Acknowledgments:** The authors extend their gratitude to the following individuals for their assistance in the field operations, treatment applications, data collection, and laboratory analysis: Tierra Freeman and Avinash Tope, Shreya Patel, Ann Freytag, Edwin Chavous, and co-operators: Ronald Bunton and Leroy Blue.

**Conflicts of Interest:** The authors declare no conflict of interest.

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
