# Peer review of "The Combined Influence of Cover Crops and Manure on Maize and Soybean Yield in a Kentucky Silt Loam Soil"

_sustainability, doi:10.3390/su11216058_

Round 1
Reviewer 1 Report
The manuscript entitled “The combined influence of cover crops and manure on maize and soybean yield in a Kentucky silt loam soil” presents a study concerning the effect of cover crop and animal manure on maize and soybean yield and biomass. The manuscript is well written and structured. The objective was stated clearly. However in my opinion some comments should be addressed before considering it to publication.
Line 18 can be one of the main (e.g. fertile topsoil erosion)
Lines 45 – 70 Authors include several examples and references regarding the benefits or changes induced by cover crops on soil properties. I miss a clear statement of the research question and the novelty of this study (see lines 66-67)
Line 71 A 4-year study where?
Line 85-86 quantitative values of max or min instead of cold, very cold… with a reference for the source data
Mean temperature varied from to please specify in which months
Line 89 How many years?
Line 90 Initial soil properties come from soils at depth? Please clarify
Line 127 Authors checked the data for normality to use parametric test?
Line 153-154 Please discuss in depth this results.
Line 172-175 What are the implications of these studies in relation to the results of the study?
Line 174 how many years Olson et al considered? Short-term 2 years as the study?
Line 186 include P value
Line 196 include a reference
Line 212 – 213 There is not enough evidence to conclude this
Line 217 – 219 This should move to the introduction to improve this part and state the research question and the context and importance / novelty of this study
Line 219 In my opinion short term referred here in this study 2 years could not be enough “to demonstrate”
Author Response
Response to Reviewer 1 Comments
Review Report (Reviewer 1)
Open Review
(x) I would not like to sign my review report
( ) I would like to sign my review report
English language and style
( ) Extensive editing of English language and style required
( ) Moderate English changes required
(x) English language and style are fine/minor spell check required
( ) I don't feel qualified to judge about the English language and style
|
Yes |
Can be improved |
Must be improved |
Not applicable |
|
|
Does the introduction provide sufficient background and include all relevant references? |
( ) |
( ) |
(x) |
( ) |
|
Is the research design appropriate? |
(x) |
( ) |
( ) |
( ) |
|
Are the methods adequately described? |
(x) |
( ) |
( ) |
( ) |
|
Are the results clearly presented? |
( ) |
(x) |
( ) |
( ) |
|
Are the conclusions supported by the results? |
( ) |
( ) |
(x) |
( ) |
Comments and Suggestions for Authors
The manuscript entitled “The combined influence of cover crops and manure on maize and soybean yield in a Kentucky silt loam soil” presents a study concerning the effect of cover crop and animal manure on maize and soybean yield and biomass. The manuscript is well written and structured. The objective was stated clearly. However in my opinion some comments should be addressed before considering it to publication.
Line 18 can be one of the main (e.g. fertile topsoil erosion)
Response. Point taken. Sentence revised as follows (L. 20)
“ Management that degrades soil can be one of the main causes of low agricultural productivity and environmental problems in many agricultural regions. “
Lines 45 – 70 Authors include several examples and references regarding the benefits or changes induced by cover crops on soil properties. I miss a clear statement of the research question and the novelty of this study (see lines 66-67)
Response. This was partially addressed in the final paragraph of the introduction (l. 78-80). It has been revised to better show the novelty and specific goal as follows:
“The number of small holder farmers embracing conservation practices is relatively small [33]. The low adoption rate could include: 1) few mechanisms to incorporate research on manure use or cover cropping practices into farm level decisions; 2) proof of short-term benefits for conservation practices on yield [34]. Our goal in this study was to show an immediate, additive effect of conservation practices that included both cover crops and animal manure on maize and soybean yield and biomass that was greater than a simple rotation effect. “
Line 71 A 4-year study where?
Response. The sentence was revised to be more specific:
“A 4-year study in Western Kentucky USA . . . “
Line 85-86 quantitative values of max or min instead of cold, very cold… with a reference for the source data. Mean temperature varied from to please specify in which months
Response. Quantitative data for winter would be of little value given that means are already reported for the site. The point being that during winter it is not cold enough for long enough to entirely shut down mineralization activities for long periods. The section has been revised as follows:
“Very cold periods in winter are usually of short duration. Mean annual temperature varied between 7 ËšC in January to 21 ËšC in July. Mean annual precipitation varied from and 991 mm in 2015 to 1524 mm in 2016 (KY mesonet data).”
Line 89 How many years?
Response: Years in prior rotation specified as follows:
“The site had been in a no-till maize-soybean rotation for approximately 10 years (personal communication with the landowner) before the start of this study.”
Line 90 Initial soil properties come from soils at depth? Please clarify
Response. Depth indicated in text and table:
“Initial soil physical, chemical, and biological properties from 0-15 cm depth are in Table 1.”
Line 127 Authors checked the data for normality to use parametric test?
Response. Yes. We added lines 152-153:
“We assessed the data for normality (P = 0.875) following the Kolmogorov-Smirnov (K-S) test.”
Line 153-154 Please discuss in depth this results.
Response. We have expanded the discussion as follows.
“The interaction of manure + cover crop for maize biomass was both positive and significant. Our initial hypothesis was that the combination of conservation practices would have an additive effect for both biomass and yield. While not purely additive, this result indicates there is benefit to cover crop and manure use together beyond their utility when used alone for maize.”
Line 172-175 What are the implications of these studies in relation to the results of the study?
Response. We have revised the paragraph as follows to provide amplification of the results:
“ The Olsen study illustrates the problem with tying benefits of conservation practices solely to yield because in the same period there were significant increase in C storage in the slightly eroded soils [32]. “
Line 174 how many years Olson et al considered? Short-term 2 years as the study?
Response. The Olsen study was a 12-year study. Noted in text (l. xx)
Line 186 include P value
Response. P value noted (l. xx).
Line 196 include a reference
Response. A reference is already included for this statement: 33. Tellatin, S.; Myers, R. Cover crops at work: Keeping nutrients out of the Waterways; USDA-SARE cover crop fact sheet series, 2018.
Line 212 – 213 There is not enough evidence to conclude this
Response. Agreed. We have amended the paragraph to indicate that our comments are speculative:
“We can speculate that because fertilization was N-based, and legumes like soybean can fix their own N, they may have received little additional benefit from the winter cover crop mineralization.”
Line 217 – 219 This should move to the introduction to improve this part and state the research question and the context and importance / novelty of this study
Response. This was a very useful suggestion and we have altered the introduction as follows to incorporate it:
“The number of small holder farmers embracing conservation practices is relatively small (Knowler and Bradshaw, 2007). The low adoption rate could include: 1) few mechanisms to incorporate research on manure use or cover cropping practices into farm level decisions; 2) proof of short term benefits for conservation practices on yield. As more producers in the US incorporate cover crops into their rotation, the expectation is they will see the benefits of this conservation practice with improvements in soil and crop performance with time. Our goal in this study was to show an immediate, additive effect of conservation practices that included both cover crops and animal manure on maize and soybean yield and biomass that was greater than a simple rotation effect. “
Line 219 In my opinion short term referred here in this study 2 years could not be enough “to demonstrate”
Response. Agreed. Sentence changed as follows.
“Soybean yields were unchanged with cover crops in the first year. However, maize yield and biomass increase was observed in the second year of a rotation that started including cover crops, which could presumably reflect improvements in soil quality and fertility. There is a further need to evaluate the effects of cover crops and manure N application rates on corn and soybean yield, and evaluation of crop and manure-specific effects.”
Reviewer 2 Report
Dear author(s),
Your paper is in good shape- it will provide valuable information to the scientific communities. Addressing the following comments will improve your quality of manuscript--
L39: incomplete sentence, for what?
L46-54: do the author(s) thinks, there could be any negative effects of cover crop on farming practices and productivity or soil quality? If so, please mention.
L56: Add more literature for manure’s advantages.
L64/65: what is ‘some’? Give a specific example.
In all figures, please change the color, so that you can show two ends of SE.
L153/154: Please re-write that last sentence of the paragraph.
Sections 3.1 and 3.2, reasoning with previously published references need to be added more to support/ oppose your findings.
Good luck!
Author Response
Response to Reviewer 2 Comments
|
Yes |
Can be improved |
Must be improved |
Not applicable |
|
|
Does the introduction provide sufficient background and include all relevant references? |
( ) |
( ) |
(x) |
( ) |
|
Is the research design appropriate? |
(x) |
( ) |
( ) |
( ) |
|
Are the methods adequately described? |
( ) |
(x) |
( ) |
( ) |
|
Are the results clearly presented? |
( ) |
( ) |
(x) |
( ) |
|
Are the conclusions supported by the results? |
( ) |
(x) |
( ) |
( ) |
Comments and Suggestions for Authors
Dear author(s),
Your paper is in good shape- it will provide valuable information to the scientific communities. Addressing the following comments will improve your quality of manuscript--
L39: incomplete sentence, for what?
Response: Revised (L. 35-37) as follows:
“Soil management, sustainability, and productivity are a global concern for food security and environmental quality [1,2].”
L46-54: do the author(s) thinks, there could be any negative effects of cover crop on farming practices and productivity or soil quality? If so, please mention.
Response: Revised (L. xx-xx) as follows:
“Additional references added for concerns raised with cover crops and immobilization, water competition, and habitat for pests.”
L56: Add more literature for manure’s advantages.
Response: Not changed. 7 references would seem to be adequate to demonstrate the beneficial properties of manure.
L64/65: what is ‘some’? Give a specific example.
Response: Revised: (L. xx-xx) as follows:
“For example, fertility recommendations from the University of Kentucky Cooperative Extension Service (University of Kentucky Cooperative Extension Service. 2018. 2018-2019 Nutrient and Lime Recommendations. AGR-1. University of Kentucky, College of Agriculture, Cooperative Extension Service, Lexington, KY. http://www2.ca.uky.edu/agcomm/pubs/agr/agr1/agr1.pdf ) give up to a 28 kg ha-1 credit to maize grown following four or fewer years of a leguminous cover crop.”
In all figures, please change the color, so that you can show two ends of SE.
Response: Open fill (with pattern) used to show the full SE bars.
L153/154: Please re-write that last sentence of the paragraph.
Response: Revised as follows:
“The interaction of manure + cover crop for maize biomass was both positive and significant.”
Sections 3.1 and 3.2, reasoning with previously published references need to be added more to support/ oppose your findings.
Response: it is to support the finding that maize responded much more strongly and positively to increased nitrogen mineralization level. The Sarr et. al., [40] is an accompanying paper currently in press.
“Prior work at this site showed a much stronger response of maize grain yield (16 Mg ha-1) than soybean (5.6 Mg ha-1) yield to mineralizable N 11.4 mg kg-1 (maize) and 9.7 mg kg-1 wk-1 [40].”
Reviewer 3 Report
Scientific Paper can be published
Round 2
Reviewer 1 Report
The manuscript has been improved by the Authors and all the comments have been addressed satisfactorily. Before considering for publication, please revise the sentence in line 88 it seems redundant “The number of small farmers … is small"
Author Response
Reviewer 2 comment:
Before considering for publication, please revise the sentence in line 88 it seems redundant “The number of small farmers … is small"
Response:
Thanks for the comment.
We revised the sentence in line 88 as follows:
The number of small-scale farmers embracing conservation practices in the US has been modest over the years [33] although an increasing number of farmers are beginning to realize the environmental benefits of these practices.
Maheteme